# Research on the Model for the Friction Coefficient of Resin-Based Friction Material and Its Experimental Verification

**DOI:** 10.3390/ma16134791

**Published:** 2023-07-03

**Authors:** Shanglong Fang, Wei Xiao, Kewen Chen, Xuding Song

**Affiliations:** 1Key Laboratory of Road Construction Technology & Equipment, Chang’an University, Ministry of Education, Xi’an 710064, China; 18363998856@163.com (S.F.); songxd@chd.edu.cn (X.S.); 2School of Architecture Engineering, Chang’an University, Xi’an 710064, China; 3Chinese Building Materials Test & Certification Group Co., Ltd., Xianyang 712021, China

**Keywords:** resin-based friction materials, friction coefficient model, wear morphology, frictional film

## Abstract

Resin-based friction materials have been widely used in the friction braking of automobiles and power machinery. Based on experiments for the variation law of friction and wear morphology, a new model for the friction coefficient of resin-based friction materials was proposed, which includes the effects of both the micro convex body on the surface of the friction material and the frictional film generated during the friction process. This quantitative model of friction coefficient materials was established for the modelling of shear strength, compressive strength, shear strength of the frictional film, contact load and wear morphology. The shear strength, compressive strength and wear morphology of the friction material were adjusted by changing the content of basalt fibers and flaky potassium magnesium titanate. Finally, the accuracy of this quantitative model of friction coefficient was verified through experiments on friction samples with different formulations and by changing the frictional contact load. The results show that the predicted friction coefficient of the model is in good agreement with the experimental friction coefficient, the difference between the upper and lower limits of the forecast is only 5.03% and 2.30%, respectively. Meanwhile, the influence of the ratio of shear strength to compressive strength on the friction coefficient is greater than the proportion of wear morphology. The proposed friction model provides a reference value for the study of new resin-based friction materials.

## 1. Introduction

Friction materials are composite materials widely used in power machinery to perform braking and transmission functions relying on friction. Friction materials are made using complex processing techniques that combine reinforcing materials, adhesives, and fillers. Among many brake friction materials, resin-based friction materials using phenolic resin as the adhesive have advantages such as ease of processing, low damage to brake discs, and low noise, and they have become the mainstream friction materials in the current automotive industry [1,2,3,4].

With the development of the transportation and mechanical industry, the requirements for brake friction materials to ensure driving safety have gradually increased in recent years [5,6,7]. The friction coefficient shows the performance of a friction material in performing braking functions, and is a key performance indicator of friction materials [4,8]. Sun et al. [9] found that the friction coefficient decreases as the content of resin matrix binder increases, which is due to the gradual decrease in strength of the frictional film, resulting in a decrease in adhesive friction. Ahmadijokani et al. [10] compared different contents of carbon-fiber-reinforced resin-based friction materials and found that as the content of carbon fiber increased, the wear resistance increased, while the friction coefficient also decreased. An amount of 4% of carbon fiber reduced the wear rate by 55%, and the friction coefficient also decreased to about 0.3. This is because the self-lubricating property of carbon fiber reduced contact stress during braking. Zhao et al. [11] prepared friction materials based on a newly designed formulation that were fabricated with different basalt fibers with different content. Other studies have shown that the mechanical strength and thermal properties of the friction materials added with basalt fiber have been improved, and the friction coefficient and wear rate have been improved [12,13,14,15,16]. Cho et al. [17] have studied the friction and wear characteristics of friction materials reinforced with potassium magnesium titanate with different morphologies, and have demonstrated that the shape of potassium titanate plays an important role in the formation of a frictional film on the friction surface. The frictional film on the surface of the friction materials has a significant impact on the friction coefficient and wear rate. The flaky potassium magnesium titanate is more conducive to the formation of a large area of frictional film on the material surface, thereby improving the friction coefficient and wear resistance of the materials [18,19,20].

In addition, the influence of frictional contact load on the friction process is also very important. The frictional contact load can affect the frictional film on the surface of friction materials and the formation of surface frictional films. When the frictional contact load is increased, the frictional film of the friction material’s surface will deform and fracture, increasing the actual contact area, leading to an increase in friction coefficient. The structure and performance of the frictional film formed on the dual surfaces under different loads are different, thereby affecting the friction and wear properties of the friction materials [21,22].

Currently, the research on friction materials mainly focuses on the qualitative analysis of formula optimization and its impact on macroscopic friction and wear properties, but the quantitative analysis of frictional properties is relatively rare [23,24]. For the contact and friction between machined surfaces, Greenwood et al. [25] assumed that the frictional film height satisfies the Gaussian distribution, and established a normal contact model (GW model) based on statistical methods. Majumdar et al. [26] characterized the surface morphology using two fractal parameters, and proposed a fractal contact model (MB model). A large number of scholars have improved these contact models [27,28,29,30,31]. However, because the contact surfaces of brake friction materials can generate debris due to severe wear, and the debris forms a frictional film with a thickness of several micrometers to tens of micrometers under complex frictional chemical reactions, which affects the friction process [32,33], the GW model and MB model only consider the impact of the surface micro convex body on the friction process, without considering the impact of the frictional film; so, they are not suitable for guiding the development of new resin-based friction materials. In this paper, a new model is proposed, which considers both the micro-convex surface of the friction material and the influence of the friction film on the friction process. The friction coefficient is quantitatively analyzed by the mechanical strength and wear morphology of the material, and the accuracy of the model is verified by experiments, which can provide reference for developing new high friction coefficient resin-based friction materials in industry.

## 2. Materials and Methods

### 2.1. Preparation of Materials

In this study, basalt fibers and flaky potassium magnesium titanate were used as the main reinforcing materials. The basalt fibers used are FX-15 basalt rock wool fibers produced by the Mineral Materials Company of Xianyang Nonmetal Mine Research and Design Institute Co., Ltd. (Xianyang, China), with a length of 109.4~493.3 μm and diameter of 5.542~6.235 μm. Potassium magnesium titanate was provided by Tangshan Yuanli New Materials Technology Co., Ltd. (Tangshan, China), with a sheet diameter of 2–5 μm, the sheet thickness was about 1 μm, and the pH value after dispersion in water was between 8 and 10. The contact of flaky potassium magnesium titanate and basalt fiber reinforcement materials in the formula can be adjusted.

The friction material samples with different formulas were prepared. The formula of the friction material is shown in Table 1. S01, S02, and S03 mainly change the content of flaky potassium magnesium titanate, while S01, S04, S05, and S06 mainly change the content of basalt fibers. Other materials in the table include the following: lightweight magnesia—3 wt.%, nitrile butadiene rubber—2 wt.%, synthetic graphite—3 wt.%, sb_2_s_3_—3 wt.%, zirconium silicate—3 wt.%, cashew oil friction powder—3 wt.%, calcined alumina—3 wt.%, petroleum coke—3 wt.%.

### 2.2. Experimental Methods

The friction material samples were prepared by hot pressing. The molded friction materials samples were obtained after mixing, pressing and heat treatment, respectively. A WDW-50 electronic universal material testing machine (Xianyang, China) was used to measure the shear strength of friction materials samples. The sample size was 15 mm × 15 mm × 6 mm. A CMT5304-30KN microcomputer (Xianyang, China)-controlled electronic universal testing machine was used to test the compressive strength of the samples. The sample size was 10 mm × 10 mm × 6 mm. Three tests were performed on each formula sample and the average was noted.

The XD-MSN constant speed friction testing machine (Xianyang, China) was used to test the friction and wear performance. The sample size was 25 mm × 25 mm × 6 mm. Each test required two samples to be pressed on the turntable of the friction testing machine, the contact load could be adjusted between 1225 and 612.5 N by weight, and the temperature during the experiment was controlled by the thermocouple automatic heating and cooling system of the testing machine. In this study, the experimental temperature was constant at 150 °C, and the friction coefficient was automatically measured by the friction testing machine. The experiment is shown in Figure 1a.

Because the frictional film adheres to a film-like structure with a thickness of only a few microns on the friction surface, it is not convenient to directly test the shear strength of the frictional film. Therefore, it is necessary to bond a metal block on the surface of the frictional film to complete the test. The experiment is shown in Figure 1b.

In Figure 1, the metal block (20 mm × 20 mm × 8 mm) is bonded to the surface of the sample that generates the frictional film through resin which is cured for 24 h to ensure that the frictional film adheres tightly to the metal block. Finally, it is installed on a WDW-50 electronic universal material testing machine to make the shear surface coincide with the friction surface. The schematic diagram of the experimental device is shown in Figure 1c. The shear strength of the friction film is calculated by recording the loading force when the metal block is separated from the friction body, and the calculation formula is
(1)$$\tau_{2}=\frac{F_{0}}{A_{0}}$$
where $\tau_{2}$ is the shear strength of the friction film, $F_{0}$ is the loading force when the metal block is separated from the friction body, and $A_{0}$ is the bonding area between the metal block and the friction material in Figure 1.

In order to control the variables, the effect of temperature on the coefficient of friction and other physical quantities were not considered in this study, leaving these topics for future research.

## 3. Results and Discussion

### 3.1. Analysis of Wear Morphology Characteristics

The friction and wear behavior of resin-based friction materials are not inherent properties of the materials, but are greatly affected by the structural properties of the frictional film at the friction interface. During the friction process, the braking pressure causes the micro convex body on the surface of the friction material to fracture and deform. The fractured debris is compacted under the action of the braking pressure and friction heat, forming a frictional film around the fractured micro convex body, thereby increasing the actual contact area [34,35].

The micro schematic diagram before and after the wear morphology of the friction material is shown in Figure 2. Figure 2a shows that the wear morphology of the friction material at the beginning of friction (<1000 revolutions); the surface is rough and full of micro convex body platforms. At this time, the flaking micro convex body platforms are being converted from debris to frictional film. The formation speed of frictional film is greater than the destruction speed, but a large area of continuous frictional film has not yet been formed, so the variation of friction coefficient is extremely unstable. Figure 2b shows the wear morphology of the friction material after being subjected to the friction disc (5000 revolutions). It can be seen that a considerable portion of the micro convex body has been damaged and deformed by the friction disc during the friction process. The wear debris formed by the micro convex body has accumulated next to the peeling micro convex body to form a continuous frictional film. The frictional film continuously adheres, shears, transfers, and re-adheres to form a continuous cycle. At this time, the area of the frictional film remains stable, so the friction coefficient reaches a stable range. In addition, even after full friction, there are still some relatively concave areas on the surface that do not participate in the friction process.

As the friction process progresses, the micro convex body is continuously sheared and peeled off to form abrasive debris, which is continuously compacted into a frictional film. The frictional film gradually grows and forms from scratch, and also constantly tears and damages under the shear action. At the initial stage of friction, the area of the film is unstable, so the friction coefficient also fluctuates greatly (Figure 3). After a period of friction, the formation speed and damage speed of the frictional film form a dynamic balance; so, the friction coefficient remains in a stable range.

### 3.2. Establish Friction Coefficient Model

The friction and wear mechanism in the above processes mainly include two parts. The first part is the peeling caused by the fracture of the micro convex body on the surface of the friction materials. The second part is the adhesive wear of the frictional film. It can be considered that the force generated in these two parts in the direction of friction is the main source of friction force. The friction coefficient model established is shown in Figure 4.

In Figure 4, the A1 part represents the micro convex body on the surface of the friction materials, the A2 part represents the frictional film formed during the friction process, and the A3 part represents a relatively concave region that is not involved in the direct friction process. The entire friction material’s surface is composed of the number (n) of regions, such as A1, A2, and A3.

Because the strength and hardness of resin-based friction materials are much lower than those of the friction disc materials (grey cast iron HT250), the main wear mechanism of part A1 is flaking along the friction direction. Part A1 is mainly destroyed and flaked unilaterally by the friction disc without destroying the surface of the friction disc. However, the scratches of the friction disc are relatively small at the macro scale. Therefore, the friction force generated by part A1 during the friction process is mainly shear stress caused by the spalling of the micro convex body. The equation for friction force is as follows (2):(2)$$F_{1}=A_{1}\cdot \tau_{1}$$
where $F_{1}$ represents the friction force generated by part A1, $A_{1}$ represents the cross-sectional area of the micro convex body, and $\tau_{1}$ represents the shear strength of the micro convex body.

The mechanism of the frictional film in part A2 is mainly adhesive wear, which is the process of shear fracture of the frictional film, i.e., the destruction and regeneration of the adhesive joint on the surface of the friction materials. Therefore, the force generated in part A2 is mainly the shear stress of the frictional film. The equation for friction force is as follows (3):(3)$$F_{2}=A_{2}\cdot \tau_{2}$$
where $F_{2}$ represents the friction force generated by the frictional film, $A_{2}$ represents the area of the frictional film, and $\tau_{2}$ represents the shear strength of the frictional film. Part A3 does not generate direct friction because it does not come into direct contact with the friction disc. The equation for the total friction force is as follows (4):(4)$$F=n\cdot (F_{1}+F_{2})$$
where F represents the total friction force and n represents the number of such areas such as A1, A2 and A3 on the surface of the material. The A1 and A2 parts are the parts that actually participate in friction, that is, the parts where the wear morphology appears on the surface of the friction sample. The following Equation (5) can be obtained:(5)$$p\cdot A=n\cdot (A_{1}+A_{2})$$
where A represents the nominal contact area of friction sample and p represents the characterization parameter of wear morphology. The larger p represents the higher proportion of wear morphology on the whole friction sample surface. Since the frictional film is formed gradually after the beginning of friction, the contact load on the surface of the friction materials is mainly borne by the part of the micro convex body A1, which results in Equation (6).
(6)$$N=n\cdot A_{1}\cdot \sigma_{1}$$
where N is contact load and $\sigma_{1}$ is the compressive strength of the compressive strength of friction materials. Equation (7) can be obtained from the definition of the friction coefficient.
(7)$$\mu =\frac{F}{N}$$

Substitute Equations (2)–(6) into Equation (7) to obtain Equation (8).
(8)$$\mu =\frac{F}{N}=\frac{n\cdot A_{1}\cdot \tau_{1}}{n\cdot A_{1}\cdot \sigma_{1}}+\frac{p\cdot A\cdot \tau_{2}}{N}-\frac{n\cdot A_{1}\cdot \tau_{2}}{n\cdot A_{1}\cdot \sigma_{1}}=\frac{\tau_{1}}{\sigma_{1}}-\frac{\tau_{2}}{\sigma_{1}}+\frac{p\cdot A\cdot \tau_{2}}{N}$$

In Equation (8), A is the nominal contact area of two friction samples, that are 2 × 25 mm × 25 mm = 1250 mm^2^, so the friction coefficient is mainly determined by the wear morphology characterization parameter p, the contact load N, the shear strength $\tau_{2}$ of the frictional film, the shear strength $\tau_{1}$ of the friction materials and the compressive strength $\sigma_{1}$. The contact load N can be directly adjusted. p, $\tau_{2}$, $\tau_{1}$ and $\sigma_{1}$ are mainly affected by the composition content of friction materials. Therefore, the content of basalt fiber and potassium magnesium titanate in the materials can be changed to adjust p, $\tau_{2}$, $\tau_{1}$ and $\sigma_{1}$. The friction coefficient model of Equation (8) will be validated in the next section.

## 4. Experimental Verification of Friction Coefficient Model

### 4.1. Shear Strength and Compressive Strength of Friction Materials

The shear strength and compressive strength tests were tested on six formula samples in Table 1, namely, friction materials with different basalt fiber content and potassium magnesium titanate content. The experimental results are shown in Figure 5. Figure 5 shows the influence curve of different composition contents on shear strength $\tau_{1}$ and compressive strength $\sigma_{1}$.

It can be seen from Figure 5 that the shear strength $\tau_{1}$ and compressive strength $\sigma_{1}$ of the friction materials are mainly influenced by basalt fibers. With the increase in basalt fiber content, the shear strength $\tau_{1}$ and compressive strength $\sigma_{1}$ reach the maximum when the basalt fibers content reaches 16%, and are 20.9 MPa and 74.6 MPa, respectively. Moreover, $\tau_{1}/\sigma_{1}$ also increases with the increase in basalt fibers content, indicating that the increase in basalt fibers on shear strength is greater. From Equation (8), it can be seen that $\tau_{1}/\sigma_{1}$ is a physical quantity that directly affects the friction coefficient of a material. This indicates that basalt fibers increase the friction coefficient mainly by increasing the material’s $\tau_{1}/\sigma_{1}$. However, the effect of flaky potassium magnesium titanate on the shear strength $\tau_{1}$ and compressive strength $\sigma_{1}$ of the friction material is small. This is because basalt fibers have a relatively large ratio and can produce a strong bridging toughening effect; so, the strength of the friction material is increased. Due to the flaky structure of potassium magnesium titanate, which lacks similar effects, the impact on various strengths of the friction material is very small.

### 4.2. Shear Strength of Frictional Film

The shear strength $\tau_{2}$ measured by the method shown in Figure 1 was tested on six formula samples in Table 1, namely, friction materials with different basalt fiber content and potassium magnesium titanate content. The experimental results are shown in Figure 6. Figure 6 shows the influence curve of composition content on shear strength $\tau_{2}$.

It can be seen from Figure 6 that the flaky potassium magnesium titanate and basalt fiber have little impact on the shear strength $\tau_{2}$ of the frictional film, which may be due to the fact that the mechanical properties of the frictional film are less affected by the reinforcement composition, and are mainly affected by the resin binder composition [9]. In addition, the frictional film is a structure that is continuously grown and destroyed during friction, with a large number of defective structures inside. Its shear strength is also greatly affected by the internal structure [22]. Moreover, in this study, because the change in composition content of flaky potassium magnesium titanate is relatively small, the impact on shear strength $\tau_{2}$ can be ignored. From the experimental results, it can be concluded that the shear strength distribution of the frictional film on the surface of the friction material reinforced by flaky potassium magnesium titanate and basalt fibers is between 0.23 and 0.25 MPa, with an average of 0.238 MPa.

### 4.3. Wear Morphology Characterization Parameter

The wear morphology characterization parameter p represents the ratio of actual contact area to nominal contact area of the friction material, where a larger p represents a larger proportion of wear morphology on the surface of the friction material. The larger the actual contact area of the same material, the greater the friction coefficient, which also conforms to Equation (8).

The reinforcing composition of flaky potassium magnesium titanate plays an important role in the formation of frictional films [18,19,20]. Figure 7 shows SEM photos of flaky potassium magnesium titanate friction samples S01, S02, and S03 with different content after friction and wear. It can be clearly seen that with the increase in flaky potassium magnesium titanate content, the proportion of wear morphology also increases, and the characterization parameter p of wear morphology also increases.

It can be seen that flaky potassium magnesium titanate has a significant impact on the area of the friction morphology. Although 8% of flaky potassium magnesium titanate samples also have obvious frictional films and spalling pits on the surface, the proportion of the material not covered by the frictional film is also relatively large. This part of the material does not have direct contact with the friction disk due to its concave surface, which causes spalling wear, nor does it cover the frictional film directly involved in friction, that is, the A3 part in the friction model (Figure 4), which has a small impact on the friction process. The larger the A3 part, the smaller the proportion of wear morphology, the smaller the characterization parameter p, and the smaller the friction coefficient. As the content of potassium magnesium titanate increases, the proportion of wear morphology becomes larger and larger, and the concave parts in the microscopic morphology are almost covered by the frictional film. The increase in the characterization parameter p may be because potassium magnesium titanate can serve as a reference point for the growth of friction films during the friction process to promote the formation of friction films on the material surface.

According to Equation (8), the friction coefficient will also increase, which is consistent with the general understanding that the larger the actual contact area, the higher the friction coefficient. It can also be seen that the flaky potassium magnesium titanate mainly improves the friction coefficient by influencing the area of the wear morphology.

### 4.4. Verifying the Influence of Friction Material Strength on Friction Coefficient

From Section 4.2, it can be seen that the shear strength $\tau_{2}$ of the frictional film is not sensitive to changes in composition. The shear strength $\tau_{2}$ can be taken as the average value of 0.238 MPa. From Section 4.3, it can be seen that the wear morphology characterization parameters p are mainly affected by the content of potassium magnesium titanate. For samples S01, S04, S05, and S06, the content of potassium magnesium titanate is 8%, and p can be considered as a constant. In Equation (8), the nominal contact area A and the contact load N can be kept constant in the experiment, and are also considered constants. At this time, the friction coefficient μ should be linearly related to $\frac{\tau_{1}}{\sigma_{1}}-\frac{\tau_{2}}{\sigma_{1}}$. Table 2 shows the measured friction coefficient values.

The shear strength $\tau_{1}$, compressive strength $\sigma_{1}$, and the ratios of the friction materials S01, S04, S05, and S06 are calculated as follows:

**Table 2 materials-16-04791-t002:** Shear strength $\tau_{1}$ and compressive strength $\sigma_{1}$ of friction materials and their ratios.

	Basalt Fiber (wt.%)	Shear Strength $\tau_{1}$ (MPa)	Compressive Strength $\sigma_{1}$ (MPa)	$\frac{\tau_{1}}{\sigma_{1}}$	$\frac{\tau_{1}}{\sigma_{1}}-\frac{\tau_{2}}{\sigma_{1}}$	μ
S01	10	13.3	56.2	0.2367	0.2324	0.404
S04	12	14.8	59.2	0.2500	0.2460	0.419
S05	14	17.1	63.3	0.2701	0.2664	0.436
S06	16	20.9	74.6	0.2802	0.2770	0.451

Figure 8 is a curve of $\frac{\tau_{1}}{\sigma_{1}}-\frac{\tau_{2}}{\sigma_{1}}$ as the abscissa and μ as the ordinate. It can be seen that μ is indeed linearly related to $\frac{\tau_{1}}{\sigma_{1}}-\frac{\tau_{2}}{\sigma_{1}}$. The maximum deviation between the fitted value and the measured value is 0.7%. Moreover, from Equation (8), it can be seen that the intersection point of the friction coefficient curve and the longitudinal axis is the numerical value of $\frac{p\cdot A\cdot \tau_{2}}{N}$. The fitted friction coefficient curve equation is Equation (9).
(9)μ=0.99x+0.173
where $x=\frac{\tau_{1}}{\sigma_{1}}-\frac{\tau_{2}}{\sigma_{1}}$, and the wear morphology characterization parameter of this formula can be calculated by substituting p = 0.712.

### 4.5. Verifying the Influence of Contact Load on Friction Coefficient

The contact load of the XD-MSN constant speed friction testing machine during friction and wear experiments can be adjusted by weights. The friction coefficient can be tested experimentally at the contact load of 1225 N, 1102.5 N, 980 N, 857.5 N, 735 N, and 612.5 N, respectively. The S01 sample was selected for this test. Because it is a sample with the same formula, the shear strength $\tau_{2}$ of the frictional film, the shear strength $\tau_{1}$ of the friction materials and the compressive strength $\sigma_{1}$ are constants. They have been measured in Section 3.1 and Section 3.2. It can be seen from Section 4.4 that the wear morphology characterization parameter p is equal to 0.712 for this formula. Moreover, according to Equation (8), it can be seen that the friction coefficient μ is inversely proportional to the contact load N. The predicted value range and measured friction coefficient value of the friction materials are shown in Figure 9.

The friction coefficient is inversely proportional to the contact load, which is also consistent with the conclusions drawn from other literature studies [21,22]. The measured friction coefficient also conforms to this rule. The measured friction coefficient is always within the predicted friction coefficient range, in which the maximum deviation of the upper limit of prediction is 5.03% and the maximum deviation of the lower limit of prediction is 2.30%. In addition, when the contact load is smaller, the measured friction coefficient is closer to the predicted lower limit. When the contact load is larger, the measured value is closer to the predicted upper limit. For example, when the contact load is 1225 N, the measured friction coefficient of the sample is 0.404, which is closer to the predicted friction coefficient when the upper limit of the shear strength of the frictional film is 0.25 MPa. When the contact load is 612.5 N, the measured friction coefficient of the sample is 0.567, which is closer to the predicted friction coefficient when the lower limit of the frictional film shear strength is 0.23 MPa. The reason for this may be that the frictional film is pressed more tightly when the pressing force is large, so the shear strength of the frictional film is closer to the predicted upper limit, and vice versa.

After experimental verification, the model of friction coefficient proposed in this study is relatively accurate in analyzing the quantitative impact of resin-based friction material composition content and contact load on the friction coefficient.

### 4.6. Importance of Various Factors on Friction Coefficient

Based on the measured friction coefficient μ, the wear morphology characterization parameter p of each sample can be calculated by substituting the measured strength data in Figure 5 and Figure 6. The values of friction coefficient μ and wear morphology characterization parameter p of samples with different potassium magnesium titanate content are shown in Table 3.

In order to compare the importance of various physical quantities on the friction coefficient, a partial derivative is obtained for each variable in Equation (8). When a physical quantity is a variable, other physical quantities are treated as constants. For example, when Equation (8) takes the partial derivative of variable $\tau_{2}$, other variables $\tau_{1}/\sigma_{1}$, p, N and A are treated as constants. In addition, based on the measured data, the numerical value range of the derivative for each variable was obtained. The ratio of material shear strength to compressive strength $\tau_{1}/\sigma_{1}$, frictional film shear strength $\tau_{2}$, wear morphology characterization parameter p and intensity of pressure N/A were derived from Equation (8). The numerical variation range after being substituted into the data is shown in Table 4.

It can be seen from Table 4 that $d(\tau_{1}/\sigma_{1})/d\mu >dp/d\mu$, so the degree of importance of the influence on the friction coefficient is $\tau_{1}/\sigma_{1}$ > p.

The intensity of pressure N/A varies widely. When the contact load N is taken as a larger value (1225 N), $d(\tau_{1}/\sigma_{1})/d\mu >dp/d\mu$, so the degree of importance of the influence on the friction coefficient is $\tau_{2}$ > N/A. When the contact load N is taken as a smaller value (612.5 N), the situation is reversed.

According to the relationship between the measured friction coefficient and the composition content of formula, it is possible to compare the importance of the influence of basalt fibers and flaky potassium magnesium titanate on the friction coefficient. The influence curve of basalt fibers and flaky potassium magnesium titanate content on the friction coefficient is shown in Figure 10.

As can be seen from Figure 10, basalt fibers have a greater impact on the friction coefficient of the materials than potassium magnesium titanate. This is because basalt fibers mainly affect the ratio of shear strength to compressive strength $\tau_{1}/\sigma_{1}$ of the materials. Flaky potassium magnesium titanate mainly affects the wear morphology characterization parameter p, which is consistent with the conclusion drawn in Table 3 that the importance of the impact on the friction coefficient $\tau_{1}/\sigma_{1}$ > p is also consistent.

## 5. Conclusions

In this study, a new model for the friction coefficient of resin-based friction materials was proposed based on experiments of the variation of friction and wear morphology of resin-based friction materials, which can quantitatively analyze the influence of friction material strength, frictional film strength, the wear morphology characterization parameter and contact load on the friction coefficient. Finally, the accuracy of the model was verified through experiments. The conclusions are as follows:

(1) The ratio of shear strength to compressive strength of friction materials, the shear strength of frictional films and wear morphology characterization parameter are positively correlated with the friction coefficient, while the contact load is negatively correlated with the friction coefficient.

(2) The importance of various factors on the friction coefficient is as follows: the ratio of shear strength to compressive strength is greater than the characteristic parameters of wear morphology. The influence of basalt fiber on the friction coefficient is greater than that of potassium magnesium titanate.

(3) Increasing the content of basalt fiber can improve the shear strength and compressive strength of resin-based friction materials and has a more obvious effect on the shear strength. Basalt fiber improves the friction coefficient of materials in a way that affects the strength of friction materials.

(4) Flaky potassium magnesium titanate is beneficial to the formation of frictional film, but has little influence on the mechanical properties of the material. Increasing the content of flaky potassium magnesium titanate will increase the proportion of wear morphology on the surface of the friction materials. Flaky potassium magnesium titanate improves the friction coefficient of the materials in a manner that affects the formation of the frictional film.

## Figures and Tables

**Figure 1 materials-16-04791-f001:**
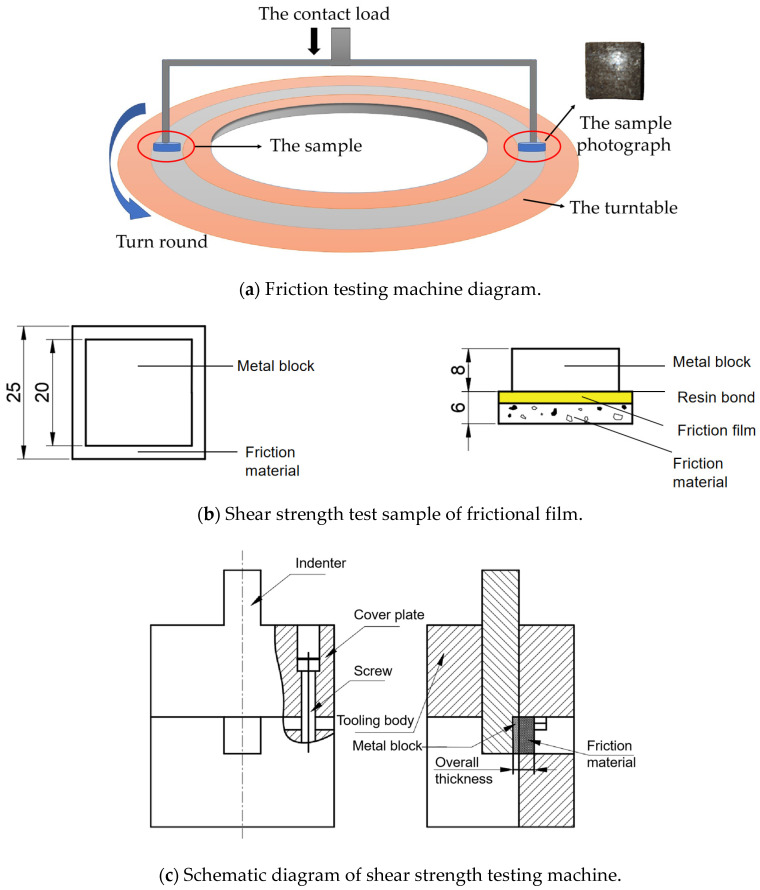
Schematic diagram of experimental equipment.

**Figure 2 materials-16-04791-f002:**
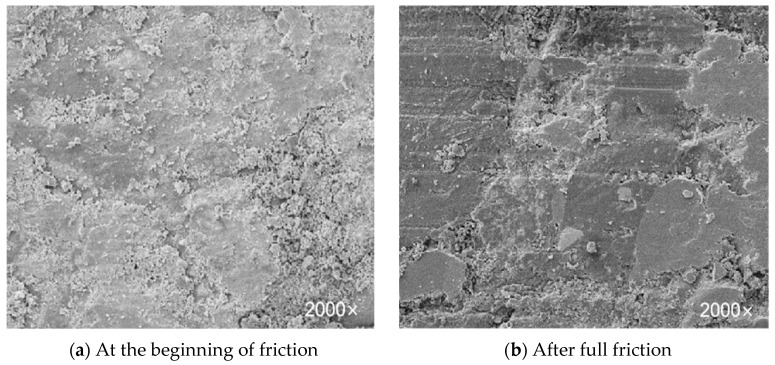
Wear morphology of friction material surface.

**Figure 3 materials-16-04791-f003:**
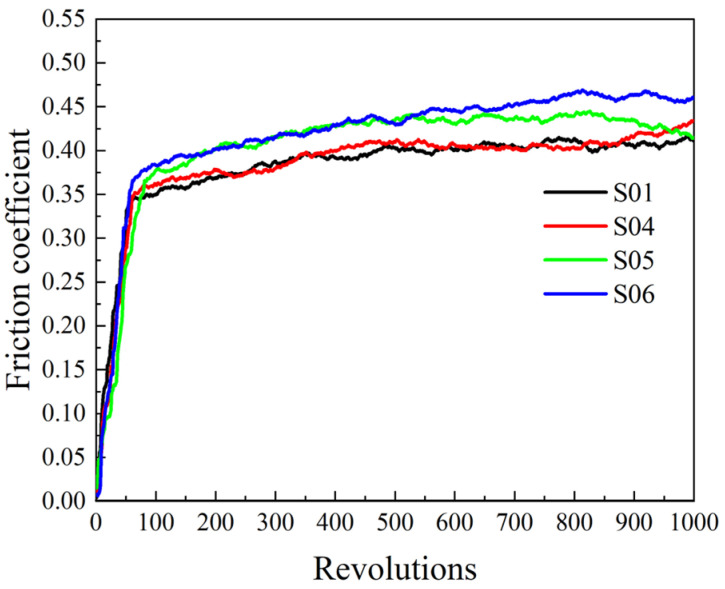
Data curve of the friction coefficient in the first 1000 revolutions with an unstable film area.

**Figure 4 materials-16-04791-f004:**
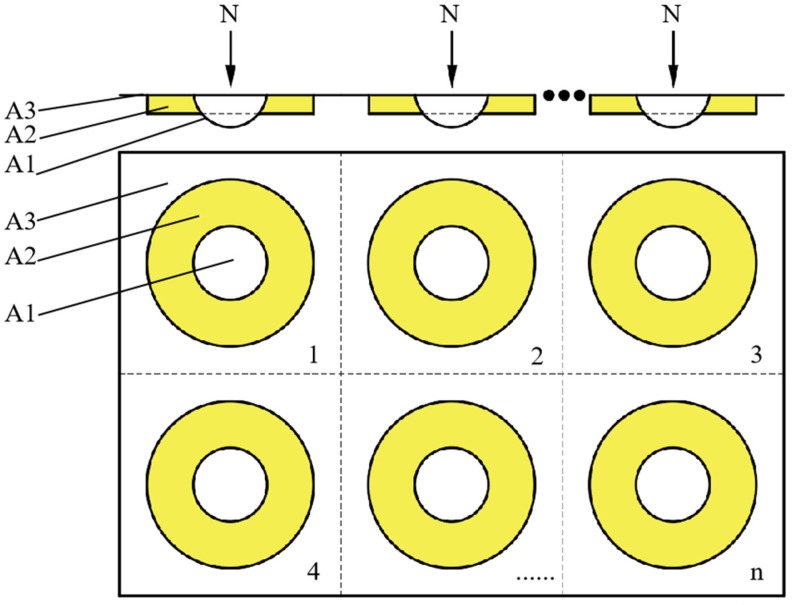
Friction coefficient model diagram.

**Figure 5 materials-16-04791-f005:**
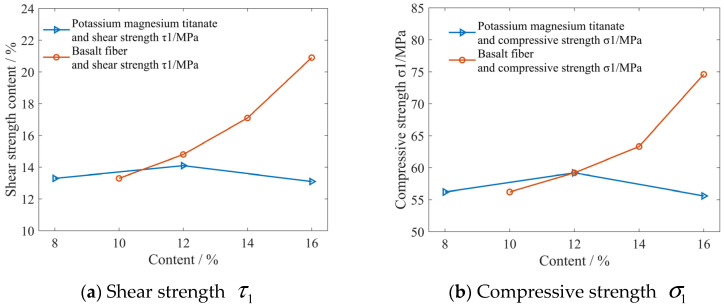
The influence curve of different composition contents on shear strength $\tau_{1}$ and compressive strength $\sigma_{1}$.

**Figure 6 materials-16-04791-f006:**
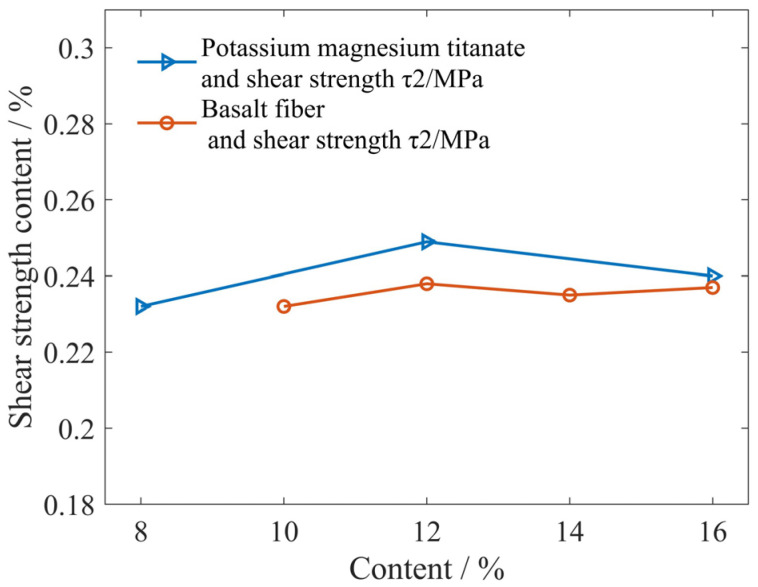
The influence curve of composition content on shear strength $\tau_{2}$.

**Figure 7 materials-16-04791-f007:**
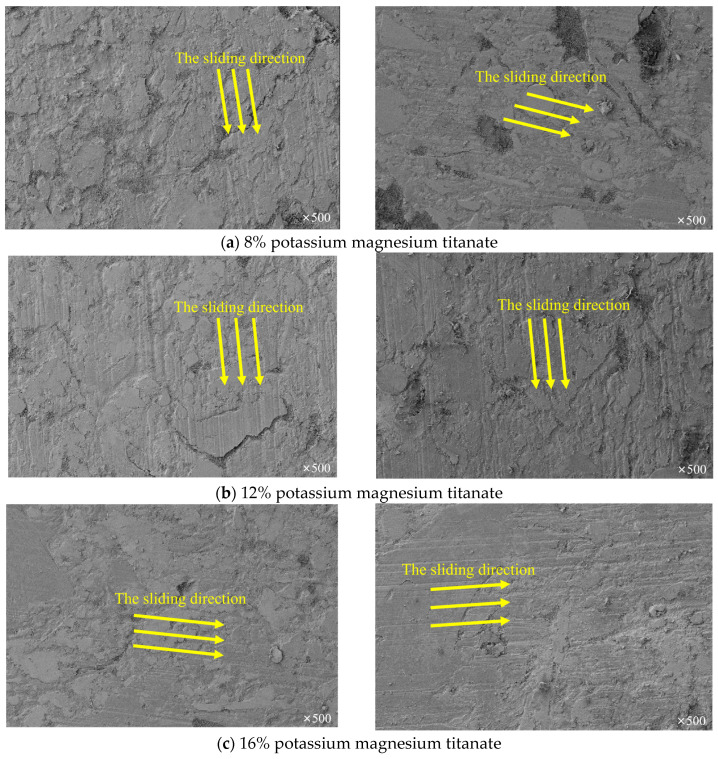
Micromorphology of materials with different potassium magnesium titanate content after wear.

**Figure 8 materials-16-04791-f008:**
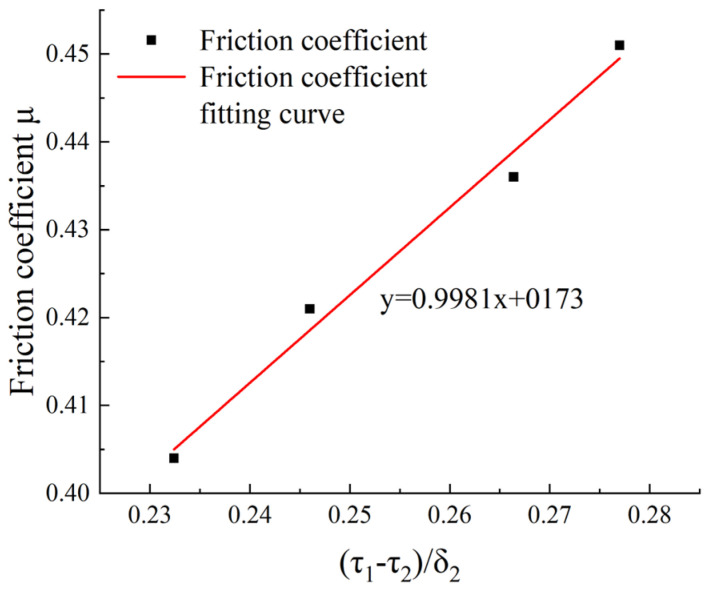
Friction coefficient fitting curve.

**Figure 9 materials-16-04791-f009:**
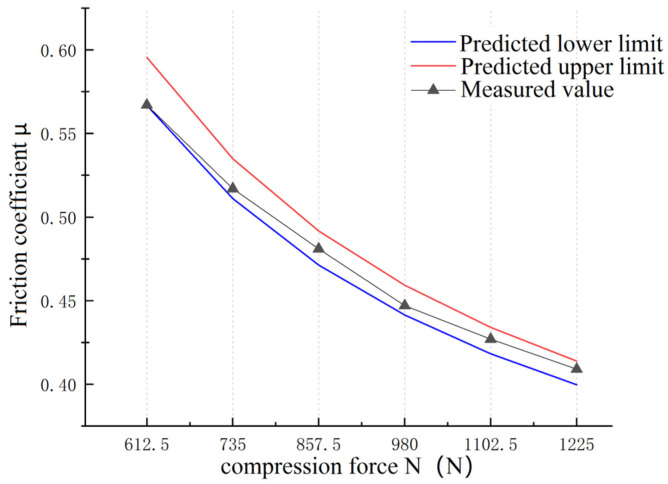
Effect of contact load on friction coefficient.

**Figure 10 materials-16-04791-f010:**
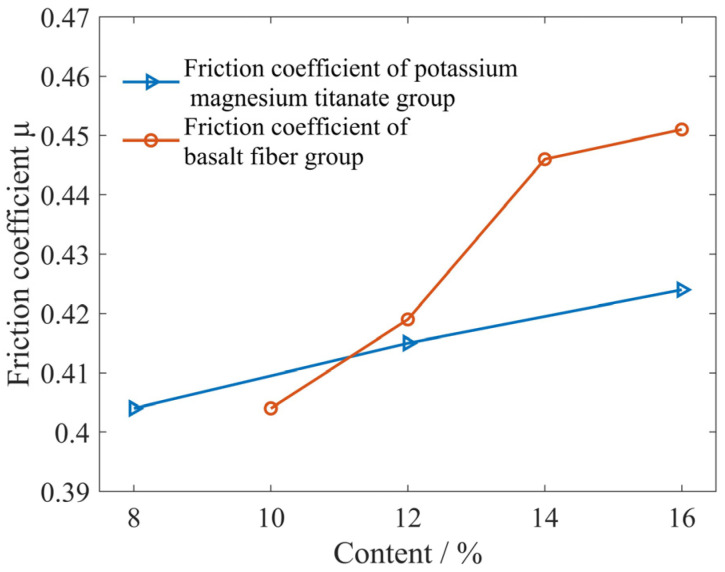
The influence curve of basalt fibers and flaky potassium magnesium titanate content on friction coefficient.

**Table 1 materials-16-04791-t001:** Formula of friction materials.

	Phenolic Resin (wt.%)	Potassium Magnesium Titanate (wt.%)	Basalt Fiber (wt.%)	Barium Sulfate (wt.%)	Hybrid Fiber (wt.%)	Other (wt.%)
S01	14	8	10	21	15	32
S02	14	12	10	17	15	32
S03	14	16	10	13	15	32
S04	14	8	12	19	15	32
S05	14	8	14	17	15	32
S06	14	8	16	15	15	32

**Table 3 materials-16-04791-t003:** Friction coefficient μ and wear morphology characterization parameter p of samples.

	Potassium Magnesium Titanate (wt.%)	μ	p
S01	8	0.404	0.712
S02	12	0.415	0.745
S02	16	0.424	0.793

**Table 4 materials-16-04791-t004:** The degree of importance for the influence on friction coefficient.

Physical Quantity	Equation Derivative	Numerical Variation Range of Equation Derivative
$\tau_{1}/\sigma_{1}$	1	1
p	$\frac{A\cdot \tau_{2}}{N}$	[0.243, 0.486]
$\tau_{2}$	$\frac{p\cdot A}{N}-\frac{1}{\sigma_{1}}$	[0.709, 1.601] MPa^−1^
N/A	$\frac{p\cdot \tau_{2}}{{\left(N/A\right)}^{2}}$	[0.178, 0.793] MPa^−1^

## Data Availability

Not applicable.

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
