# Peer review of "Research on the Model for the Friction Coefficient of Resin-Based Friction Material and Its Experimental Verification"

_materials, 2023, doi:10.3390/ma16134791_

Round 1

Reviewer 1 Report

In this study, a model has been proposed for the coefficient of friction of resin-based friction materials, taking into account both the surface micro-convexity of the friction materials and the effect of the friction film on the friction process. The accuracy of the model has been verified through experiments. The article can be accepted when revision is made taking into account the following recommendations.

1-Figure 1 should be a little more detailed. In addition, the friction testing machine should be given in figure 1 and the position of the load of the sample should be shown.

2- It should also be noted that the dimensions given in Figure 1 are in mm.

3-In page 4. "The shear strength of the frictional film is calculated using the experimental values when  the metal block and the friction body separate." Details such as how the shear strength is calculated and how the amount of applied force is determined should be specified.

4- In page 4: "...so the variation of friction coefficient is extremely unstable" Only one optical image given in Figure 3. How is it understood from this image that the friction coefficient is unstable?

5- In my opinion there is no need to use stress symbols in header 4.1

6-It would be more useful to show the sliding direction in Figure 7.

7- It is recommended to use additional images related to production details in the "Preparation of Materials" section given in Section 2.1.

Author Response

Dear reviewers:

We are very grateful to your comments for the manuscript. According with your advice, we tried our best to amend the relevant part and made some changes in the manuscript. All of your questions were answered below. You can find details in the attachment. To make the reply more visible, “Point” represents questions raised by reviewers, and “Response” are our answers for these questions.

We appreciate for Reviewers’ warm work earnestly, and hope that the correction will meet with approval.

Once again, thank you very much for your comments and suggestions.

Yours Sincerely,

Wei Xiao

Reviewer 2 Report

The authors proposed a friction mode for resin-based friction material. They validated their model based on several shear strength measurements. Overall, the language written is poor, resulting in the difficulty to understand the manuscript. The abstract needs to be rewritten to better highlight the important findings of the study. More importantly, the friction model presented is too superficial and lacks in-depth fundamental understanding. Below are other comments related to the manuscript that the authors can refer to:

Comment #1:

Please be consistent with the use of “coefficient of friction”. The term is interchangeably used with “friction coefficient”. 

Comment #2:

In the abstract, please revise the use of the terms “positively correlated” and “negatively correlated”. Keep it simple as to whether it is correlated or no.

When mention about correlation, highlight the statistical parameters that indicate how well the correlation is. E.g., r-squared? 

Comment #3:

In the abstract, please provide quantitative comparison between the model and the experiment. E.g., how many percent deviation?

Comment #4:

In the abstract, please explain clearly what is meaning of “importance of various factors on the friction coefficient”? Suggest revising the statement.

Comment #5:

Please update the in-text referencing format. Cite only the last name of the authors. Also, check the names of cited authors. There are mistakes as well. E.g., “Majumda” should be “Majumdar”.

The list of references also has a lot of mistakes. Please update reference list accordingly. 

Comment #6:

The introduction lacks emphasis on the motivation of the study. The study gap is not clearly explained. 

Comment #7:

The friction model is not clearly explained. It is difficult to gauge how the model can correlate to the experiment. E.g., How does the parameters in the friction model measured experimentally? 

Comment #8:

At the end of the study, the authors used on the nominal contact area of the friction sample, which does not follow the typical observations of a rough surface contact. This also makes the study less valuable as it only requires the shear strength as the input to the model. Such approach is commonly reported in literature. 

Comment #9:

I suggest the authors to focus on the experimental data and determine the relationship between the shear strength of the materials rather than attempting to propose a friction model. The depth of the fundamental knowledge to derive a friction model is not sufficient at this stage.

Extensive language editing is required. 

Author Response

(The authors gave the same response as above.)

Reviewer 3 Report

The manuscript with a title “Research on the model for friction coefficient of resin-based friction material and its experimental verification” proposes and verifies a new model for friction coefficient of resin-based friction materials based on the wear morphology characteristics, as well as the micro convex body and frictional film effects. The manuscript provides some useful and relevant information and insights for the design and optimization of resin-based friction materials.

However, the manuscript also has some limitations and weaknesses that need to be addressed and improved. Here are some specific recommendations and suggestions for each section of the manuscript:

Introduction

1.       The aims are not clearly stated. The last paragraph provides some background information and literature review, but it does not explicitly state what the main research question or objective is.

2.       The authors should cite more recent publications (from the past five years) that are relevant to their topic and show the current state of the art in the field.

Materials and methods

3.       The authors need to provide more details about the materials used.

4.       The authors need to provide more details about the experimental procedures. They should also explain how they ensured the accuracy, precision, and reproducibility of their experiments.

Results and Discussion

5.       The authors must ensure that all symbols are explained in the description of the formulas.

6.       The captions of Figures 3 to 9 must be improved for clarity and accuracy.

Experimental verification of friction coefficient model

7.       The authors should use consistent and clear symbols and units for the physical quantities and their derivatives.

Conclusions

8.       The authors should explain how their study contributes to the existing knowledge or practice in the field of resin-based friction materials. They should also mention some limitations or challenges of their study and suggest some directions or recommendations for future research. This would help to show the significance and relevance of their study.

The manuscript needs major revision before it can be considered for publication. The authors need to address all the recommendations and suggestions mentioned above in order to improve their manuscript.

Author Response

(The authors gave the same response as above.)

Reviewer 4 Report

1. The image, table, equations arrangements should be done with more care.

please check the minor mistakes.

Author Response

(The authors gave the same response as above.)

Round 2

Reviewer 2 Report

the authors have responded to the comments in an acceptable manner.

The language is readable but improving it would help make it easier to understand the content.

Reviewer 3 Report

Accept in present form